# The Next Frontier in ART: Harnessing the Uterine Immune Profile for Improved Performance

**DOI:** 10.3390/ijms241411322

**Published:** 2023-07-11

**Authors:** Nathalie Lédée, Marie Petitbarat, Laura Prat-Ellenberg, Géraldine Dray, Virginie Vaucoret, Alaa Kazhalawi, André Rodriguez-Pozo, Nada Habeichi, Lea Ruoso, Nino Guy Cassuto, Mona Rahmati

**Affiliations:** 1MatriceLab Innove Laboratory, Immeuble Les Gemeaux, 2 Rue Antoine Etex, 94000 Creteil, France; marie.petitbarat@matricelabinnove.com (M.P.);; 2Centre d’Assistance Médicale à la Procréation, Hôpital des Bluets, 4 Rue Lasson, 75012 Paris, France; laura.prat-ellenberg@bluets.org (L.P.-E.);; 3Laboratoire Drouot, 21 Rue Drouot, 75010 Paris, France; 4London Women’s Clinic, 113-115 Harley Street, London W1G 6AP, UK; mona.rahmati@londonwomensclinic.com

**Keywords:** uterine immune profiling, assisted reproductive technology, endometrium, cytokines, personalized medicine, implantation failures

## Abstract

Assisted reproduction techniques have improved considerably in recent decades, but despite these advances, success rates remain relatively low. Endometrial immune profiling involves the analysis of cytokine biomarkers in the endometrium during the mid-luteal phase. This profiling aims to provide insights into the immune environment of the uterus. The aim is to identify immune disturbances and thus guide the development of personalized therapeutic approaches. The first part of the review looks back at the emergence of innovative concepts, highlighting the specificity of the human uterine environment at the time of implantation. Based on this new knowledge, biomarkers have been selected for endometrial immune profiling. The second part details the results of clinical studies conducted over the last ten years. These clinical results suggest that this approach can increase the rate of live births in patients suffering from repeated implantation failures or repeated pregnancy loss. Uterine immune profiling represents a clinical innovation that can significantly improve the performance of medically assisted reproduction treatments through personalized strategies tailored to the local immune profile. Innovation in personalized medicine for assisted reproduction is crucial to improving the success rates of fertility treatments, while reducing the risks and costs associated with ineffective or unnecessary interventions.

## 1. Introduction

Assisted reproductive technologies (ART) have significantly improved in recent decades and have become a widely accepted treatment option for infertility. However, despite the advancements in ART, the success rates still remain relatively low. The live birth rate per initiated treatment cycle reported in 2018 is approximately 30% for women under 35 years old and decreases drastically with age [1]. Moreover, ART is often associated with emotional, psychological, and financial stress, particularly when facing repeated failure of implantation (RIF) or recurrent pregnancy loss (RPL) [2]. RIF and RPL can also have significant social and economic consequences, including loss of productivity, increased healthcare costs, and a decline in the quality of life [3].

Although ART has significantly improved the chances of achieving a pregnancy for patients struggling with infertility, there is still a need for continued research and innovation to improve the success rates and reduce the emotional and financial burden associated with treatment. It is estimated that 15% of couples trying for natural conception, representing 186 million people, suffer from infertility worldwide [4].

Endometrial immune profiling involves the analysis of immune cells and biomarkers in the endometrium (inner lining of the uterus) to understand the local immune status and its potential impact on fertility and pregnancy. This approach can help to identify immune disturbances as potential contributors to infertility or pregnancy loss and guide the development of personalized therapeutic approaches [5].

In assisted reproductive medicine, endometrial immune profiling may be particularly useful for women who have experienced RPL [6] or RIF despite multiple cycles of in vitro fertilization (IVF) [7]. By identifying specific biomarkers, the endometrial immune profiling can guide the selection of appropriate immune-modulating therapies to improve pregnancy outcomes.

Overall, this innovative strategy aims to develop more personalized and effective approaches to assisted reproductive medicine, with a focus on identifying and addressing potential immune disturbances contributing to infertility and pregnancy loss [8].

## 2. Emergence of the Concept: The Uterine-Specific Immune Environment

### 2.1. Animal Models

Several reasons explain why the uterine immunity has been historically neglected in assisted reproductive technologies (ART). First, in the early days of ART, much of the research and clinical attention were directed towards embryonic factors that could affect implantation and pregnancy success, such as embryo quality and morphology. On the other hand, the immune system is complex, and there was a general lack of knowledge regarding its functions, especially in the context of pregnancy [9]. Additionally, immune cells can be highly dynamic and difficult to isolate and characterize. However, in recent years, there has been an increasing recognition of the importance of the uterine immunology in ART, and there is at present a growing body of research in this area.

In 1996, Y. S. Loke and his colleagues at the University of Cambridge identified a specific subset of immune cells called the uterine natural killer (uNK) cells, which are found in high numbers in the endometrium during the implantation window [10]. They showed the important role of these cells in the implantation and the establishment of a healthy pregnancy. Specifically, they found that the uNK cells produce cytokines and growth factors that help to promote the growth of blood vessels in the uterus and support the developing embryo [11]. They also discovered that the uNK cells are regulated by hormones and that abnormal levels of uNK cells may contribute to implantation failure and recurrent pregnancy loss in mice. This research helped to shift the focus of reproductive immunology from the adaptive immune system to the innate immune system and the importance of uNK cells in the establishment of pregnancy [12].

Studies conducted by T. Wegmann and G. Chaouat using abortion-prone mouse models have made significant contributions to our understanding of uterine immunity and the role of cytokines in successful pregnancy [13].

Crucial concepts related to the immune environment of the uterus and its role in pregnancy were highlighted in their research:

The immune privilege of the uterus and the presence of specific uterine natural killer (uNK) cells were demonstrated, promoting maternal tolerance towards the fetus [14].

The concept of immunotrophism was introduced, emphasizing the role of the immune system in regulating the growth and development of the conceptus [15,16].

The discovery was made that a shift towards a Th2-dominant immune environment is necessary for successful implantation and the maintenance of pregnancy [13]. Later, it was further revealed that a Th2 dominance is crucial, but the presence of Th1 cytokines is also required [17].

These early discoveries set the stage for further investigations regarding the role of the immune system in infertility and assisted reproductive medicine. With the emergence of new technologies and research methods, such as flow cytometry and gene expression analysis, researchers were able to more closely examine the various immune cells and molecules present in the uterine environment. This led to the identification of a network of cytokines, chemokines, and other factors that play important roles in embryo implantation and pregnancy success [18].

### 2.2. Translation to Human

Human implantation is a complex process involving the synchronized interaction of the embryo and the endometrium [19]. Compared to other mammalian species, human implantation is characterized by several unique features, such as the maternal local preparation through the decidualization of the endometrium, programmed deep and extensive invasion, and immunological tolerance.

Human implantation can be described as a three-stage process, comprising apposition, adhesion and the invasion phase, followed by placentation proper. During the apposition and adhesion phase, the blastocyst approaches the receptive endometrium and makes contact with the luminal surface. This initial contact is mediated by specific molecules on the surface of both the blastocyst and the endometrium. The blastocyst expresses various adhesion molecules, including integrins and selectins, while the endometrium expresses their respective ligands [20]. During this phase, the endometrium undergoes various changes to prepare for implantation, including an increase in vascular permeability and secretory activity, which provide nutrients and other factors to support the developing embryo. The apposition and adhesion phase is critical for a successful implantation and sets the stage for subsequent invasion of the endometrium by the blastocyst.

The adhesion phase of implantation is often considered as a pseudo-inflammatory step because it involves the interaction of the blastocyst with the endometrial surface, leading to the activation of inflammatory-like responses. This phase is characterized by the initial interaction between the blastocyst and the receptive endometrium, which is mediated by the binding of specific adhesion molecules on the surface of the blastocyst and the endometrial epithelium. This interaction triggers a cascade of events, including the release of cytokines and chemokines, which ultimately lead to the attachment of the blastocyst to the endometrium [21]. The inflammatory-like response during this phase helps to prepare the invasion phase of human implantation. One of the particularities of human invasion is the formation of specialized structures called endovascular extravillous trophoblasts (EVTs), which invade the lumen of the spiral arteries and replace the endothelial cells. This process is regulated by various factors, including cytokines, growth factors, and adhesion molecules.

Human implantation occurs approximately 6–7 days after ovulation. The window of implantation (WOI) is a crucial time frame when the endometrium undergoes changes in response to hormonal signals from the ovary, preparing itself to receive and support an embryo. During the WOI, the endometrium becomes highly vascularized and enriched with nutrients and growth factors, allowing it to support the implantation and early growth of the embryo [22,23]. Uterine immune cells play a critical role in the process, as they contribute to the establishment of a receptive environment for the embryo to implant and develop [24,25].

Several studies conducted during the WOI show that there is a switch from an adaptive immunity to an innate immunity that takes place in the endometrium. There is a decrease in the number and function of B and T lymphocytes in the endometrium, and an increase in the number and function of uNK cells, macrophages, and dendritic cells. The presence of these innate immune cells is essential to promote implantation, establish pregnancy, and support placental development [26]. This shift is essential to create an immunologically tolerant environment for the developing embryo, which is considered as a semi-allograft, meaning that it carries antigens from the father that are not present in the mother. It allows the developing embryo to evade rejection by the maternal immune system and promotes the establishment of an immunologically tolerant environment for the developing fetus. Locally, the shift towards a TH-2-dominant immune response is thought to play a crucial role in establishing immune tolerance towards the developing embryo. This is achieved by promoting the differentiation of immune cells towards a regulatory phenotype, which promotes tissue remodeling, angiogenesis, and the suppression of pro-inflammatory responses. Studies have compared the immune profiles of uterine and circulating immune cells during implantation, showing significant differences between the two compartments [27]. Uterine immune cells, including uNK cells, macrophages [28], dendritic cells [29], and T regulatory cells [30], have been shown to be highly specialized and functionally distinct from their circulating counterparts. For example, uNK cells are highly abundant during early pregnancy and have a unique phenotype and function that differs from circulating NK cells [27,31]. They secrete cytokines and growth factors that support embryo implantation and placentation, and they play a crucial role in the regulation of trophoblast invasion and vascular remodeling [26].

The uterine environment during the WOI characterized by a Th2-dominant cytokine profile promotes the development and the adequate expression and differentiation of immune cells, such as uNK cells, dendritic cells, macrophages, and T regulatory cells, which are required to promote embryo implantation, trophoblast invasion, and placental development [24].

The uNK cells, for example, produce several cytokines and growth factors that facilitate trophoblast invasion and angiogenesis [32,33]. Dendritic cells and macrophages help in promoting an immune-tolerant environment by inducing the development of T regulatory cells, which suppress immune responses and promote immune tolerance [29,34]. The T regulatory cells also contribute to the maintenance of a Th2-dominant cytokine profile by secreting anti-inflammatory cytokines [35].

Thus, the balance between Th1 and Th2 cytokines plays a critical role in the success of implantation, and a Th2-dominant cytokine environment is essential for promoting immune tolerance and embryo implantation.

During implantation, the invading trophoblast cells of the embryo remodel the spiral arteries in the maternal decidua to allow for increased blood flow to the developing placenta [36]. This remodeling process involves the replacement of the endothelial and smooth muscle cells of the spiral arteries with fetal trophoblast cells, leading to a low-resistance, high-capacity blood supply to the developing fetus [37].

Hence, the destabilization of spiral arteries before their invasion is an important step in the endometrial preparation for human implantation.

The inadequate remodeling of spiral arteries can occur due to various factors, including immune dysregulation, oxidative stress, and genetic factors. These factors can affect the behavior and function of the cells involved in the vascular remodeling process, such as trophoblast cells and uNK cells, leading to an insufficient or incomplete remodeling. The precise mechanisms behind the failure of spiral artery remodeling are still not fully understood, but it is thought to involve aberrant immune responses and an inadequate production of cytokines and growth factors.

### 2.3. Selecting Key Immune Targets to Define the Uterine Immune Profile

The objective is to understand how the endometrium is prepared for an effective implantation and to detect the imbalances able to impair the process of implantation.

To reach such objective, we quantified during the mid-luteal phase the RNA expression, through RT-qPCR, of some biomarkers selected not for their specificity (there is a wide redundancy of cytokines and growth factors locally), but for the key information they provide on the local TH-2/TH-1 balance, the destabilization of spiral arteries, and uNK mobilization and maturation.

The ratio of IL-18/TWEAK mRNA is a biomarker that serves as an indicator of both angiogenesis and Th1/Th2 balance. IL-18/TWEAK provides insights into the local immune environment and the potential presence of an immune deviation towards Th1 cytokines, which can affect the implantation process. On the other hand, IL-15/Fn-14 mRNA is used as a biomarker to assess the activation and maturation status of uterine natural killer (uNK) cells, along with the evaluation of uNK-CD56 cell count.

#### 2.3.1. IL-18

Interleukin-18 (IL-18) is a pro-inflammatory cytokine that plays a key role in the innate immune system, known to play a role in various physiological processes, including inflammation, immune response, and tissue remodeling [38]. In the context of reproduction, IL-18 has been found to be involved in embryo implantation and placental development [39]. In the context of pregnancy, IL-18 intervention was identified in various aspects of placentation, such as trophoblast invasion and differentiation, angiogenesis, and immune regulation [40,41,42].

Studies have shown that IL-18 is expressed in the endometrium during the WOI and is involved in the regulation of trophoblast invasion and migration. IL-18 also plays a role in the modulation of uNK cells activity in the endometrium, which is critical for successful implantation and the maintenance of pregnancy [43]. In addition, IL-18 promotes angiogenesis and vascularization in the placenta, which is important for the exchange of nutrients and oxygen between the mother and the developing fetus. However, the excessive or dysregulated expression of IL-18 can have negative effects on pregnancy outcomes, such as preterm birth, preeclampsia, and fetal growth restriction [44].

IL-18 has a bivalent role in the context of the TH-1/TH-2 paradigm. On the one hand, IL-18 is known to promote TH-1 immune responses, which are characterized by the activation of cytotoxic T cells and the production of proinflammatory cytokines, such as interferon-gamma (IFN-γ) and tumor necrosis factor-alpha (TNF-α) [45]. On the other hand, IL-18 has also been shown to have some TH-2-like activities, such as the stimulation of eosinophils and the production of IL-13 and IL-5. The bivalent nature of IL-18 is thought to be related to its ability to interact with different cytokines and to modulate the immune response in a context-dependent manner [46]. For example, IL-18 can synergize with IL-12 to promote TH-1 responses, but it can also synergize with IL-4 to enhance TH-2 responses. Additionally, the effects of IL-18 may also depend on the timing and context of its expression, as well as on the presence of other cytokines and immune cells in the microenvironment.

#### 2.3.2. IL-15

Interleukin-15 (IL-15) is a cytokine that plays an important role in the immune system, as well as in reproductive processes, such as embryo implantation and placentation [45,47,48,49]. IL-15 is known to promote the survival, proliferation, and maturation of immune cells, such as uNK cells, and to promote the production of other cytokines, such as IL-6 and TNF-alpha, which play a role in implantation and placental development [50].

### 2.4. TWEAK and Fn-14

TWEAK (TNF-like weak inducer of apoptosis) is a type II transmembrane protein that binds to its receptor, Fn-14 (fibroblast growth factor-inducible 14), which is expressed on a variety of cells, including uNK cells [51,52]. TWEAK and Fn-14 interaction has been implicated in various physiological and pathological processes, including embryonic development, angiogenesis, inflammation, and apoptosis [53].

TWEAK, as a member of the TNF superfamily, shares similarities with TNF-α. However, TWEAK has more diverse and sometimes opposing roles in different physiological and pathological processes. TWEAK and its receptor Fn-14 signaling regulate the cytotoxicity of uterine NK cells, which is crucial for controlling trophoblast invasion and preventing fetal rejection [51,54,55]. TWEAK has also been shown to regulate the expression of other cytokines, such as IL-15 and IL-18, which are involved in the regulation of uNK cell cytotoxicity and survival. Thus, TWEAK appears to be an important local immunoregulator of cytotoxicity during implantation and early pregnancy.

### 2.5. CD56

CD56 is an important marker of uNK cells and is used to identify and quantify the mobilization of these cells in the endometrium during the WOI. Although useful to assess the recruitment of the uNK cells, the documentation of the immune environment of these cells seems more important. The evaluation of their mobilization alone, from our perspective, offered poor information.

## 3. From Concept to Clinical Studies

### 3.1. The Endometrial Diagnosis

The objective of the uterine immune profiling is to document whether the endometrium is prepared for an effective adhesion and controlled invasion when the embryo is transferred. If the immune profile is dysregulated, to regulate the immunological profile, some personalized recommendations are provided to the patient. The physician must ensure the integrity of the cavity before proceeding with the immune profiling of the uterus [56]. The thickness of the endometrium must be appropriate, and the absence of chronic endometritis (CD138, MUM1) must have been verified or, if applicable, treated with antibiotics before any immunological work-up is conducted [57]. To identify the uterine immune profile, it is necessary to explore the endometrial immune environment since these unique local reactions cannot be reflected by a blood test. In order to assess this local immune state, endometrial samples are collected during the mid-luteal phase, by a pipelle biopsy, and RT-qPCR is used to explore the mRNA expression of key-selected cytokines: interleukin-18, interleukin-15, TWEAK, Fn-14, and CD56 [58].

This method of immunological endometrial profiling has been patented as a technique to increase implantation success in assisted fertilization (PCT/EP2013/065355). The expression of each biomarker is normalized to the mean expression of reference genes, which allows the identification of an immune profile for each patient.

To establish the endometrial immune profile, a step-by-step procedure was applied, considering first the IL-18/TWEAK mRNA (reflecting the local angiogenesis and possibly a Th1 deviation), then the CD56+ cell count (reflecting the uNK cell mobilization), and finally the IL-15/Fn-14 mRNA (indicative of uNK cell maturation and uNK cytotoxic activation).

Endometrial immune profiles can be classified into four types:

A balanced endometrial immune profile, which is characterized by IL-18/TWEAK and IL-15/Fn-14 mRNA and a CD56+ cell count in the same range as previously defined in the fertile control cohort.

An underactive endometrial immune profile, which is defined by low mRNA ratios for IL-15/Fn-14 (reflecting immature uNK cells) and/or IL-18/TWEAK, as well as a limited uNK recruitment.

An overactive endometrial immune profile is characterized by elevated mRNA ratios of IL-18/TWEAK and/or IL-15/Fn-14, along with a high CD56+ cell count.

A mixed endometrial immune profile, which is defined by a high mRNA IL-18/TWEAK (suggesting an excess of Th-1 cytokines) and a low IL-15/Fn-14 (reflecting immature uNK cells).

The balanced profile suggests that the endometrium is ready to experience the following steps of implantation, which are the apposition, adhesion, and invasion.

In the underactive profile, the endometrium may not be fully ready for adhesion and promoting adequate immunotrophism during initial placentation.

In the two later profiles (overactive and mixed profile), the endometrium may be in a state that is able to reject the embryo and be prepared for the crucial step of trophoblast invasion.

### 3.2. Suggestion of Personalization

After the endometrial collection and analysis, the physician would receive the type of immune profile identified and a suggested treatment plan based on the endometrial testing. These suggestions are organized into six sections based on the immune profile.

(a)Endometrial scratching is recommended for cases with low IL-15/Fn-14, indicating uNK cell immaturity, in order to promote uNK cell maturation [59]. The scratching procedure is typically performed during the mid-luteal phase of the preceding cycle to induce the expression of chemokines, adhesion molecules, and innate immune cells [60].(b)In cases where the local IL-18 expression is documented as low, it has been suggested to avoid exposure to high concentrations of estrogens, such as those induced by ovarian hyperstimulation during IVF cycles [61,62].(c)Glucocorticoid supplementation is recommended as a first-line treatment to reduce Th-1 cytokines, decrease uNK cytotoxicity, and alleviate hyperactivation in lymphokine-activated killer cells for patients with overactivated and mixed immune profiles [59,62]. In cases where corticoid treatment is not effective, low-molecular-weight heparin (LMWH) was considered as an alternative due to its well-documented anti-complement effect [63,64]. As a second line of treatment, the intravenous slow perfusion of Intralipid^®^ was suggested to control the hyperactivation of NK cells and to regulate a Th-1-predominant cytokine balance [65,66,67]. Only a test under therapy, showing the normalization of the endometrial profile under the suggested medication, would attest of its efficacy.(d)For overactivated and mixed profiles, the hormonal adaptation of the luteal phase is recommended. This involved the use of high daily doses of vaginal progesterone (1200 mg) or a dual route of administration, such as vaginal and oral or vaginal and subcutaneous, to take advantage of the immunosuppressive properties of progesterone [68,69]. In cases of the elevated expression of IL-18, oral estradiol supplementation at a dose of 4 mg is recommended to downregulate its levels [61,62]. Progesterone influences the maternal immune system through multiple pathways and some hormonal receptors may modulate subsequent immune events [70]. It induces the production of PIBF (progesterone-induced blocking factor) to inhibit NK cell activity and promotes the production of galectin-1, which supports the development of tolerogenic dendritic cells, which in turn will induce the expansion of IL-10-secreting regulatory T cells [71].(e)The supplementation of the luteal phase with human chorionic gonadotrophin (hCG) is also one the suggested therapeutic options. During the mid-luteal phase, we recommended hCG supplementation in cases of low CD56 mobilization or immaturity of uNK cells. Previous studies have demonstrated that hCG triggers the maturation and proliferation of uNK cells, while promoting uterine angiogenesis [72,73,74,75]. HCG is naturally produced by the embryo and is directly involved in the local reaction by inducing an adequate angiogenesis while controlling the activation of uNK cells at the maternal–fetal interface.(f)In specific cases, sexual intercourse after embryo transfer is recommended. Seminal plasma has been found to have a beneficial effect on the endometrium, as it induces the expression of pro-inflammatory cytokines and chemokines and recruits immune cells [75]. Therefore, we suggest sexual intercourse in cases where the endometrial immune activation is low. However, we do not recommend exposure to seminal plasma in cases of overactivated or mixed immune profiles.

The suggested personalized treatments aiming to counteract the local imbalance, if diagnosed, are summarized in Table 1.

### 3.3. Extended Clinical Cohort Studies in Populations Who Would Benefit in Theory of Personalized Strategy

Clinical studies: a step-by-step demonstration. Although a medical innovation may seem well-documented and attractive, its utility can only be demonstrated through constructed clinical studies. In our field, the timing of implantation has been identified as a key feature in human reproduction, and various tests based on transcriptomic gene selection have been developed to determine the optimal day for embryo transfer. One such leading tests was the Endometrial Receptivity Array (ERA, Igenomix), which has been used for ten years. However, its effectiveness is currently being discussed and debated due to conflicting reports from clinical studies [76,77].

Patients with a history of RIF and patients with a history of RPL are the two populations who seem to benefit from uterine immune profiling to increase their chance of achieving a successful pregnancy [6,7,78].

#### 3.3.1. Context of Repeated Implantation Failure (RIF)

Between 2012 and 2018, two cohorts were conducted among patients with a history of unexplained RIF. The first cohort included 324 patients, while the second cohort included 1145 patients, with a median range of IVF attempts of 3.5 in both cohorts and a median range of nine embryos replaced without any pregnancy [7,78]. As reported by ESHRE in 2018, the live birth rate (LBR) at the third attempt was 20.3% and 17.2% at the fourth attempt [1]. The endometrial immune profiles of these patients were analyzed, and a dysregulation was found in 81.7% and 82.8% of the patients in the first and second cohorts, respectively. In the first cohort study, overactivation was diagnosed in 56.6% of cases, while low activation was observed in 25%. In the second cohort study, overactivation was diagnosed in 57% of cases, while low activation was observed in 23%. The LBR at the first subsequent embryo transfer for dysregulated and subsequently treated patients was significantly higher at 39.8% and 38.4% in the first and second cohort studies, respectively, compared to patients with no dysregulation, for which no treatment could be offered, whose LBR was significantly lower at 19.4% and 26.9%.

#### 3.3.2. Context of Recurrent Pregnancy Loss (RPL)

Recurrent pregnancy loss, defined as at least three consecutive miscarriages following spontaneous pregnancies, occurs in 1–2% of couples trying to conceive [79,80].

In a large prospective cohort study conducted in 2020, 164 patients with unexplained RPL were included. According to the uterine immune profiling test, among these patients, 80.5% had a uterine immune dysregulation, 23.5% had a local immune under-activation, 45% had an over-immune activation, and 12% had a mixed immune profile. Personalized care based on the identified deregulation was associated with a significantly higher ongoing pregnancy rate compared to non-dysregulated patients (38.4% vs. 26.9%, *p* < 0.002) [7].

In another 2021 study, 104 patients with RPL were analyzed retrospectively [6]. Patients initially underwent a standard extensive recurrent pregnancy loss (RPL) screening, and if any anomalies were identified, they were corrected prior to undergoing an endometrial biopsy for uterine immune profiling. A minimum follow-up period of 6 months was conducted with personalized care provided as indicated based on the comprehensive assessment. Success was defined as achieving a live birth after following the recommended treatment plan, while failure was defined as either not achieving pregnancy or experiencing a subsequent miscarriage despite undergoing the targeted therapies. Out of the 104 patients included in the study, 75% were diagnosed with an endometrial immune dysregulation. Among these patients, 31% had an under-active uterine immune profile, 50% had an overactive immune profile, and 19% had a mixed pattern. Uterine immune profiling was found to be significantly associated with a higher live birth rate (LBR) when a dysregulation was identified and treated accordingly (55% vs. 45%, *p* = 0.01). Conversely, an absence of local dysregulation (indicating a seemingly balanced immune environment) was associated with a higher risk of recurrent pregnancy loss (RPL), suggesting the need to identify other underlying causes. Dysregulated immune profiles were significantly associated with three times higher LBR compared to non-dysregulated profiles (OR = 3.4, 95% CI 1.27–9.84), and five times higher when an overactive profile was treated with immunotherapy (OR = 5, 95% CI 1.65–16.5). These two studies suggest that 75 to 80% of patients with unexplained RPL may benefit from personalized therapy based on their uterine immune profile.

#### 3.3.3. Controlled Cohort Study

In controlled cohort studies, 193 patients with RIF underwent endometrial immune profiling, and personalized treatments were administered to those diagnosed with immune dysregulation [81]. These patients were matched to a control group of 193 RIF patients who did not undergo endometrial immune profiling. Among the analyzed group, 78% had a uterine immune dysregulation and received personalized care. Their corresponding live birth rate was significantly higher than the control group (30.5% vs. 16.6%, OR: 2.2 [1.27–3.83], *p* = 0.004), with a simultaneous drastic reduction in miscarriages per initiated pregnancy (17.9% vs. 43.2%, OR: 0.29 [0.12–0.71], *p* = 0.005). The 22% of analyzed patients who had no dysregulation did not differ from their matched controls for live birth rate and miscarriages.

Randomized controlled trials are a crucial step for validating the clinical efficiency of any innovation. However, new innovations in assisted reproductive technology (ART) often skip this step, mainly due to feasibility issues. Patients with history of RIF or RPL are reluctant to participate to randomized studies because of their painful past experiences. To overcome this problem, we included good prognosis IVF/ICSI patients, who are not the classical population typically benefitting from immune profiling. The objective of the PRECONCEPTIO trial (NCT02262117) was to evaluate the impact of personalized care based on the endometrial immune dysregulation type on subsequent LBR. From October 2015 to February 2023, we prospectively enrolled 492 patients. These patients underwent immune endometrial profiling, and if a dysregulation was diagnosed, they were randomized to either a conventional or personalized embryo transfer and followed until birth. Our hypothesis was that personalized care would increase the birth rate from 25% to 40% per embryo transfer if a deregulation had been diagnosed and the care personalized accordingly. Out of the 492 patients who were included, 483 were successfully analyzed. Of these, 107 endometrial immune profiles (22.3%) were not dysregulated, while 375 (77.7%) were dysregulated, with 147 (30%) having an under-active immune profile and 229 (47%) having an overactive or mixed profile. Among the patients with a dysregulated endometrium, 189 were randomized to receive conventional treatment and 187 to receive a personalized treatment. During the trial, 21% of the included patients did not continue the protocol due to various reasons, such as loss of follow-up, discontinuation of the protocol, no embryo to transfer, or spontaneous pregnancy. To date, 157 patients have been successfully completed an embryo transfer in the conventional group, 145 in the personalized group, and 86 in the non-dysregulated group. Follow-up is ongoing, and results will be available soon. The first surprising result of this trial is that endometrial dysregulation is not specific to patients with RIF and RPL, but concerns everyone with distinct impacts based on maternal age and embryo quality. The embryo and its environment are theoretically equipped to trigger adhesion to the endometrium through the secretion of pro-adhesive molecules, as demonstrated with MUC-1, but can also secrete immunosuppressive molecules. The embryo is therefore the first immunoregulator, but if it fails, diagnosing the uterine immune profile and correcting the endometrial dysregulation involved may help.

### 3.4. Understanding Immunotherapy for an Effective Precise Medicine in IVF

#### 3.4.1. The Efficacy of Immunotherapy Needs to Be Tested

Regarding therapy and personalized medicine, it is crucial to recognize that a single approach cannot be universally applied. Cochrane reviews and meta-analyses of immunotherapy, such as glucocorticoids (GC), low-molecular-weight heparin (LMWH), and mechanical procedures, such as endometrial scratching, have shown limited efficacy in IVF [82,83,84,85,86]. The lack of effectiveness in these treatments stems from the fact that they are not tailored to a precise molecular diagnosis, but rather rely on a general context of infertility [60]. There are scientists who hold the viewpoint that the efficacy of many immunological therapies has not been sufficiently demonstrated and that these treatments are often based on unproven assumptions regarding the need for the down-regulation of natural killer (NK) cell activity [87].

Uterine and decidual natural killer (NK) cells have often been perceived only as a potential threat to the conceptus, while their crucial role as a supportive and protective factor is frequently overlooked [88]. However, it is essential to justify each prescription based on the observed dysregulation in the patient’s endometrial immune profile. The effectiveness of a drug is determined by its ability to restore balance to a dysregulated immune profile when administered as a therapy. For women with endometrial overactive or mixed immune profile, GC are recommended as first-line treatment. This is because they have been reported to decrease the levels of Th-1 cytokines, NK cytotoxicity, and the hyperactivation of lymphokine-activated killer cells. Additionally, the administration of therapy aims to modulate the Th1/Th2 balance when it is skewed towards Th1 cytokines [89,90,91,92]. LMWH (low-molecular-weight heparin) is considered as an alternative option in cases where corticosteroids are not effective. This is because LMWH has a well-documented anti-complement effect, which can help to address resistance to corticosteroid treatment [64,93].

In a study conducted in 2018, the endometrial immune profiles of 55 patients with RIF who were initially classified as having immune over-activation were evaluated. The patients underwent treatment with GC to assess the rate of normalization of their initial immune profiles [94]. The results of the study showed that under GC treatment, immune biomarkers were normalized in 54.5% of the RIF patients initially classified as having immune over-activation. However, it was observed that, in 29.1% of the cases, there was a counterintuitive negative increase in immune biomarkers, and in 16.5% of the cases, there was only partial normalization. These findings suggest that testing the sensitivity to GC in RIF patients with immunological dysregulation could be valuable. It is important to note that less than half of the RIF patients may be responders to GC treatment, indicating the need for personalized approaches and further investigation to identify patients who are likely to benefit from GC therapy. Intralipid^®^ (IL) therapy is considered as a potential second-line treatment option for women with immune overactive profiles who have not been able to conceive with glucocorticoid (GC) therapy or whose immune profile remains dysregulated despite GC treatment. The exact mechanism by which IL exerts its immunomodulatory effects is not yet fully understood. However, it is believed that IL has the ability to inhibit the production of pro-inflammatory mediators, particularly Th1 cells, and possesses immunosuppressive properties on natural killer (NK) cells. This therapy aims to restore immune balance and create a more favorable environment for embryo implantation and pregnancy. Further research is needed to elucidate the precise mechanisms and evaluate the efficacy of IL therapy in improving reproductive outcomes for women with immune dysregulation [66,95].

In a study involving 108 patients with a history of unexplained RIF or RPL, endometrial profiling was conducted before and under Intralipid (IL) therapy prior to their next embryo transfer. The objective of administering IL through slow perfusion was to regulate the observed immune over-activation in the endometrium [67]. Among the patients who achieved successful pregnancy with IL therapy and had undergone sensitivity testing before embryo transfer, a significant decrease was observed in the levels of three biomarkers used to diagnose immune over-activation: CD56 cell count, IL-18/TWEAK, and IL-15/FN-14. Among the tested patients, 27% showed resistance to IL therapy. However, for those who responded positively to IL treatment, the live birth rate (LBR) was excellent, reaching 55%. These findings suggest that IL therapy has the potential to effectively modulate the endometrial immune profile and improve pregnancy outcomes in patients with immune dysregulation.

#### 3.4.2. The Endometrial Scratching

Endometrial scratching is a procedure that highlights the importance of understanding the immunological basis guiding a personalized treatment approach, as it directly affects the immune response during implantation [60]. The biological rationale behind the procedure of endometrial scratching is to enhance the expression of adhesion molecules during the subsequent mid-luteal phase [59,96]. However, it should be noted that profiles of low local activation are observed in only a subset of recurrent implantation failure (RIF) patients. Specifically, these profiles are found in approximately 33% of RIF patients, which corresponds to 384 out of a total of 1145 patients [7]. When the procedure of endometrial scratching is performed specifically in patients with documented endometrial immune under-activation, we observed a significant improvement in the ongoing pregnancy rate. In this subgroup of patients, the ongoing pregnancy rate reached 38.5% at the subsequent embryo transfer, with a total of 181 out of 384 patients achieving pregnancy. These results highlight the importance of targeting the procedure to patients with specific immune profiles, as blindly applying endometrial scratching without considering the immune status of the endometrium can have negative outcomes or even be potentially harmful, particularly in cases of endometrial immune over-activation. It is crucial to perform endometrial scratching at the appropriate time, triggering the expected immune reaction, such as during the mid-luteal phase or in a carefully timed manner, in order to optimize the chances of successful implantation and pregnancy [86,97]. Gnainsky et al. (2010) [59] previously reported that performing a local injury during the mid-luteal phase leads to modifications in endometrial expression during the subsequent luteal phase. Specifically, they observed changes in the recruitment of macrophages and dendritic cells, as well as the expression of adhesion molecules [55]. Similarly, Liang et al. [56] found that endometrial scratching performed in the mid-luteal phase promotes the local production of vascular endothelial growth factor (VEGF) during the subsequent mid-luteal phase [94].

## 4. Conclusions

The immune state of the endometrium has been a neglected factor in reproductive medicine and infertility management. However, the uterine immune profiling represents a clinical innovation that can significantly enhance ART performance through personalized strategies tailored to the local immune profile. Clinical studies, both conducted and ongoing, suggest that this approach can increase LBR while reducing time-to-child. Therefore, it is time to avoid the blind prescription of immune modulators and instead diagnose the need for them and verify their efficacy through testing. Personalized medicine taking in account the uterine side in reproductive medicine involves tailoring treatments based on individual patient characteristics, such as the uterine immune profile, to increase the chances of a successful pregnancy. Innovation in the field of personalized medicine in ART is crucial as it can significantly improve the success rates of fertility treatments while reducing the risks and costs associated with ineffective or unnecessary interventions.

## Figures and Tables

**Table 1 ijms-24-11322-t001:** Suggestions in function of the immune profile documented.

Suggestion of Personalization	Endometrial Immune Profile			
	No Dysregulation	Under Active Immune Profile	Overactive Immune Profile	Mixed Immune Profile
Endometrial scratching (mid-luteal phase preceding the transfer)	No	Yes	No	No or Yes (therapy test)
Higher dosage estrogens	No impact	No	Yes	No
Immunotherapy (GC, LMWH, IL)	No	No	Yes (therapy test)	Yes (therapy test)
Higher progestative luteal support	No	No	Yes	Yes
Luteal hCG supplementation	No	Yes	No	No or Yes (therapy test)
Exposure to seminal plasma	No impact	Yes	No	No

Evaluation of the endometrium under treatment is called as therapy test. It assesses the normalization (or otherwise) of the profile in response to immunotherapy in the case of overactivated and mixed profiles, and will guide the physician in the possible addition of luteal scratching/HCG support in the case of mixed profiles, if necessary. GC: Glucocorticoids, LMWH: Low molecular weight heparin, IL: slow perfusion of intralipids).

## Data Availability

All the full-text of our researches are available on the site https://matricelabinnove.com/recherche-et-development/, accessed on 1 June 2023.

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
