# Peer review of "The Next Frontier in ART: Harnessing the Uterine Immune Profile for Improved Performance"

_ijms, 2023, doi:10.3390/ijms241411322_

Round 1

Reviewer 1 Report

Dear Authors,

I read your article with interest. The immunological features showed are quite exhaustive and this should be a comprehensive review very useful.

I have some minor comments:

1) I would cite also the role of chronic endometritis in impairing embryo implantation. Although recently inserted in literature, its role is quite undebatable in affection embryo implantation. Especially, you mentioned uNK CD 156, but I think also CD 138 should be mentioned. Similarly, the role of MUM-1 should be taken into account.

2) Similarly, seeing other studies, the importance of progesterone and estrogen receptors modulation (or resistance) is pabulum for further consideration on the role of immunological features in repetitive implantation failure.

3) Table 1 needs more details and should be rendered more accurate (e.g. the legend).

Good

Author Response

We thanks the two reviewers for their comments and their suggestions

Modifications of the manuscript has been done accordingly. All the sentences previously used in our own publications were modified to avoid any repetition as required by the editors

R.1 I would cite also the role of chronic endometritis in impairing embryo implantation. Although recently inserted in literature, its role is quite undebatable in affection embryo implantation. Especially, you mentioned uNK CD 156, but I think also CD 138 should be mentioned. Similarly, the role of MUM-1 should be taken into account.

I totally agree. We therefore add a paragraph specifying that endometrial thickness, integrity of the cavity and detection of chronic endometritis with CD138 or MUM-1 immunostaining should have been done (with adequate treatment if any) before any uterine immune profiling.

Similarly, seeing other studies, the importance of progesterone and estrogen receptors modulation (or resistance) is pabulum for further consideration on the role of immunological features in repetitive implantation failure.

Regarding selective progesterone and estrogens receptor modulation, we add some references in the section dedicated to the hormonal adaptation- Thank you

Table 1 needs more details and should be rendered more accurate (e.g. the legend): Table 1 has been modified

We hope the manuscript will be acceptable in this present form

Nathalie Lédée

Reviewer 2 Report

This manuscript is a review that aims to improve pregnancy success in ART using the uterine immune profile. The review initially covers the literature on "the uterine specific immune environment," which is well-written. To enhance flow, the authors should consider combining the three items starting with "they" in to a single paragraph or using bulleted points. Additionally, citations 35 and 36 should be re-ordered in the References, and subsequent citations should be re-numbered accordingly. The statement "which is later than on other species" is false and should be removed. Throughout the manuscript. there are several short paragraphs that could be combined for better readability. In Section 1.3. the title "immune" should be lowercase as it should be in the Abstract. The use of the term ratio is redundant.  The second section covers "From concept to clinical studies." In section 2.1, it is suggested that the last paragraph be moved to lead into the first sentence. In section 2.2, the table should follow the first mention of it. Additionally, "The six..." should be moved to the start of the list of profiles.

Manuscript is well-written. The authors should review usage of which and that: which precedes non-essential clauses and that precedes essential clauses. if the clause can be dropped without affecting the meaning of the sentence use which.

Author Response

We thanks the two reviewers for their comments and their suggestions

Modifications of the manuscript has been done accordingly. All the sentences previously used in our own publications were modified to avoid any repetition as required by the editors

To enhance flow, the authors should consider combining the three items starting with "they" in to a single paragraph or using bulleted points: Modification done

citations 35 and 36 should be re-ordered in the References, and subsequent citations should be re-numbered accordingly. : References were reorganized accordingly to their number

The statement "which is later than on other species" is false and should be removed.: It has been removed. Thanks for this important correction

Throughout the manuscript. there are several short paragraphs that could be combined for better readability; Done

 In Section 1.3. the title "immune" should be lowercase as it should be in the Abstract: done

 The use of the term ratio is redundant: we deleted some

 The second section covers "From concept to clinical studies." In section 2.1, it is suggested that the last paragraph be moved to lead into the first sentence: Done- thank you it’s clearer

In section 2.2, the table should follow the first mention of it. Additionally, "The six..." should be moved to the start of the list of profiles; we modified the place of the table

We hope the manuscript will be acceptable in this present form

Nathalie Lédée
